# Mapping single-cell atlases throughout Metazoa unravels cell type evolution

**Alexander J Tarashansky[1], Jacob M Musser[2†], Margarita Khariton[1†], Pengyang Li[1], Detlev Arendt[2,3], Stephen R Quake[1,4,5], Bo Wang[1,6]\***

[1]Department of Bioengineering, Stanford University, Stanford, United States; [2]European Molecular Biology Laboratory, Developmental Biology Unit, Heidelberg, Germany; [3]Centre for Organismal Studies, University of Heidelberg, Heidelberg, Germany; [4]Department of Applied Physics, Stanford University, Stanford, United States; [5]Chan Zuckerberg Biohub, San Francisco, United States; [6]Department of Developmental Biology, Stanford University School of Medicine, Stanford, United States

**Abstract** Comparing single-cell transcriptomic atlases from diverse organisms can elucidate the origins of cellular diversity and assist the annotation of new cell atlases. Yet, comparison between distant relatives is hindered by complex gene histories and diversifications in expression programs. Previously, we introduced the self-assembling manifold (SAM) algorithm to robustly reconstruct manifolds from single-cell data (Tarashansky et al., 2019). Here, we build on SAM to map cell atlas manifolds across species. This new method, SAMap, identifies homologous cell types with shared expression programs across distant species within phyla, even in complex examples where homologous tissues emerge from distinct germ layers. SAMap also finds many genes with more similar expression to their paralogs than their orthologs, suggesting paralog substitution may be more common in evolution than previously appreciated. Lastly, comparing species across animal phyla, spanning sponge to mouse, reveals ancient contractile and stem cell families, which may have arisen early in animal evolution.

**\*For correspondence:**
wangbo@stanford.edu

[†]These authors contributed equally to this work

**Competing interests:** The authors declare that no competing interests exist.

## Introduction

There is much ongoing success in producing single-cell transcriptomic atlases to investigate the cell type diversity within individual organisms (*Regev et al., 2017*). With the growing diversity of cell atlases across the tree of life (*Briggs et al., 2018*; *Cao et al., 2019*; *Fincher et al., 2018*; *Hu et al., 2020*; *Musser et al., 2019*; *Plass et al., 2018*; *Siebert et al., 2019*; *Wagner et al., 2018*), a new frontier is emerging: the use of cross-species cell type comparisons to unravel the origins of cellular diversity and uncover species-specific cellular innovations (*Arendt et al., 2019*; *Shafer, 2019*). Further, these comparisons promise to accelerate cell type annotation and discovery by transferring knowledge from well-studied model organisms to under-characterized animals.

However, recent comparative single-cell analyses are mostly limited to species within the same phylum (*Baron et al., 2016*; *Geirsdottir et al., 2019*; *Sebé-Pedrós et al., 2018*; *Tosches et al., 2018*). Comparisons across longer evolutionary distances and across phyla are challenging for two major reasons. First, gene regulatory programs diversify during evolution, diminishing the similarities in cell-type-specific gene expression patterns. Second, complex gene evolutionary history causes distantly related organisms to share few one-to-one gene orthologs (*Nehrt et al., 2011*), which are often relied upon for comparative studies (*Briggs et al., 2018*; *Shafer, 2019*). This effect is compounded by the growing evidence suggesting that paralogs may be more functionally similar than orthologs across species, due to differential gain (neo-functionalization), loss (non-functionalization),

or partitioning (sub-functionalization) events among paralogs (*Nehrt et al., 2011*; *Prince and Pickett, 2002*; *Stamboulian et al., 2020*; *Studer and Robinson-Rechavi, 2009*).

Here, we present the Self-Assembling Manifold mapping (SAMap) algorithm to enable mapping single-cell transcriptomes between phylogenetically remote species. SAMap relaxes the constraints imposed by sequence orthology, using expression similarity between mapped cells to infer the relative contributions of homologous genes, which in turn refines the cell type mapping. In addition, SAMap uses a graph-based data integration technique to identify reciprocally connected cell types across species with greater robustness than previous single-cell data integration methods (*Haghverdi et al., 2018*; *Hie et al., 2019*; *Polański et al., 2019*; *Stuart et al., 2019*).

Using SAMap, we compared seven whole-body cell atlases from species spanning animal phylogeny, which have divergent transcriptomes and complex molecular homologies (*Figure 1A–B* and *Supplementary file 1*). We began with well-characterized cell types in developing frog and zebrafish embryos. We found broad concordance between transcriptomic signatures and ontogenetic relationships, which validated our mapping results, yet also detected striking examples of homologous cell types emerging from different germ layers. We next extended the comparison to animals from the same phylum but with highly divergent body plans, using a planarian flatworm and a parasitic blood fluke, and found one-to-one homologies even between cell subtypes. Comparing all seven species from sponge to mouse, we identified densely interconnected cell type families broadly shared across animals, including contractile and stem cells, along with their respective gene expression programs. Lastly, we noticed that homologous cell types often exhibit differential expression of orthologs and similar expression of paralogs, suggesting that the substitution and swapping of paralogs in cell types may be more common in evolution than previously appreciated. Overall, our study represents an important step toward analyzing the evolutionary origins of specialized cell types and their associated gene expression programs in animals.

## Results

### The SAMap algorithm

SAMap iterates between two modules. The first module constructs a gene-gene bipartite graph with cross-species edges connecting homologous gene pairs, initially weighted by protein sequence similarity (*Figure 1C*). In the second module, SAMap uses the gene-gene graph to project the two single-cell transcriptomic datasets into a joint, lower-dimensional manifold representation, from which each cell's mutual cross-species neighbors are linked to stitch the cell atlases together (*Figure 1D*). Then, using the joint manifold, the expression correlations between homologous genes are computed and used to reweight the edges in the gene-gene homology graph in order to relax SAMap's initial dependence on sequence similarity. The new homology graph is used as input to the subsequent iteration of SAMap, and the algorithm continues until convergence, defined as when the cross-species mapping does not significantly change between iterations (*Figure 1E*).

This algorithm overcomes several challenges inherent to mapping single-cell transcriptomes between distantly related species. First, complex gene evolutionary history often results in many-to-many homologies with convoluted functional relationships (*Briggs et al., 2018*; *Nehrt et al., 2011*). SAMap accounts for this by using the full homology graph to project each dataset into both its own and its partner's respective principal component (PC) spaces, constructed by the SAM algorithm, which we previously developed to robustly and sensitively identify cell types (*Tarashansky et al., 2019*). The resulting within- and cross-species projections are concatenated to form the joint space. For the cross-species projections, we translate each species' features into those of its partner, with the expression for individual genes imputed as the weighted average of their homologs specified in the gene-gene bipartite graph. Iteratively refining the homology graph to only include positively correlated gene pairs prunes the many-to-many homologies to only include genes that are expressed in the same mapped cell types.

Second, frequent gene losses and the acquisitions of new genes result in many cell type gene expression signatures being species-specific, limiting the amount of information that is comparable across species. Restricting the analysis of each dataset to only include genes that are shared across

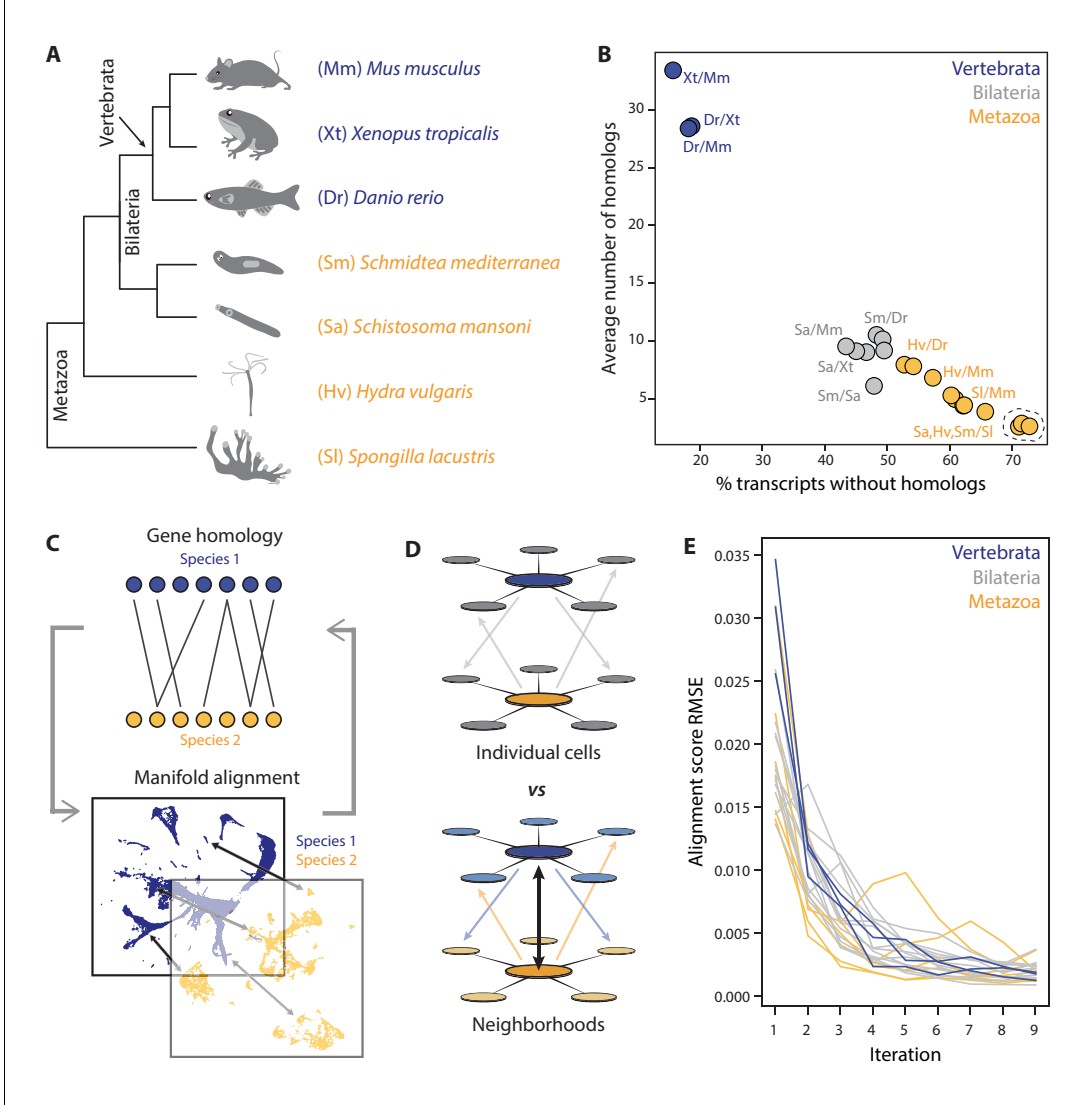

**Figure 1.** SAMap addresses challenges in mapping cell atlases of distantly related species. (**A**) Schematic showing the phylogenetic relationships among seven species analyzed. (**B**) Challenges in mapping single-cell transcriptomes. Gene duplications cause large numbers of homologs per gene, determined by reciprocal BLAST (cut-off: E-value $<10^{-6}$), and frequent gene losses and the acquisition of new genes result in large fractions of transcriptomes lacking homology, which limits the amount of information comparable across species. (**C**) SAMap workflow. Homologous gene pairs initially weighted by protein sequence similarity are used to align the manifolds, low dimensional representations of the cell atlases. Gene-gene correlations calculated from the aligned manifolds are used to update the edge weights in the bipartite graph, which are then used to improve manifold alignment. (**D**) Mutual nearest neighborhoods improve the detection of cross-species mutual nearest neighbors by connecting cells that target one other's within-species neighborhoods. (**E**) Convergence of SAMap is evaluated by the root mean square error (RMSE) of the alignment scores between mapped clusters in adjacent iterations for all 21 pairwise comparisons of the seven species.

The online version of this article includes the following figure supplement(s) for figure 1:

**Figure supplement 1.** Scalability of SAMap.

---

species would result in a decreased ability to resolve cell types and subtypes with many species-specific gene signatures. SAMap solves this problem by constructing the joint space through the concatenation of within- and cross-species projections, thus encoding all genes from both species.

Third, the evolution of expression programs gradually diminishes the similarity between homologous cell types. To account for this effect, SAMap links cell types across species while tolerating their differences. Cells are mapped by calculating each of their $k$ mutual nearest cross-species neighbors

in the combined projection. To establish more robust mutual connectivity, we integrate information from each cell's local, within-species neighborhood (*Figure 1D*), overcoming the inherent stochasticity of cross-species correlations. Two cells are thus defined as mutual nearest cross-species neighbors when their respective neighborhoods have mutual connectivity. It is important to note that the magnitude of connections is not directly calculated from their expression similarity, allowing cell types with diverged expression profiles to be tightly linked if they are among each other's closest cross-species neighbors.

Lastly, SAMap is robust to technical batch effects between datasets that are collected through different platforms. For instance, we have succeeded in running SAMap on datasets containing hundreds of thousands of cells that were collected with different single-cell platforms, including 10X genomics, Drop-Seq, SmartSeq, and MARS-Seq. SAMap runtimes were typically less than an hour on an average desktop computer for the largest dataset we tested (*Figure 1—figure supplement 1*). Further, SAMap overcomes potential memory issues when running on large datasets by chunking its computationally intensive operations into smaller blocks, saturating the memory usage with respect to the number of cells (*Figure 1—figure supplement 1B*).

## Homologous cell types emerging from distinct germ layers in frog and zebrafish

We first applied SAMap to the *Xenopus* and zebrafish atlases, which both encompass embryogenesis until early organogenesis (*Briggs et al., 2018*; *Wagner et al., 2018*). Previous analysis linked cell types between these two organisms by matching ontogeny, thereby providing a reference for comparison. SAMap produced a combined manifold with a high degree of cross-species alignment while maintaining high resolution for distinguishing cell types in each species (*Figure 2A*). We measured the mapping strength between cell types by calculating an alignment score (edge width in *Figure 2B* and color map in *Figure 2C*), defined as the average number of mutual nearest cross-species neighbors of each cell relative to the maximum possible number of neighbors.

SAMap revealed broad agreement between transcriptomic similarity and developmental ontogeny, linking 26 out of 27 expected pairs based on previous annotations (*Figure 2B* and *Supplementary file 2*; *Briggs et al., 2018*). The only exception is the embryonic kidney (pronephric duct/mesenchyme), potentially indicating that their gene expression programs have significantly diverged. In addition, SAMap succeeded in drawing parallels between the development of homologous cell types and matched time points along several cell lineages (*Figure 2C*). While the concordance was consistent across cell types, we noticed that the exact progression of developmental timing can vary, suggesting that SAMap can quantify heterochrony with cell type resolution.

SAMap also linked a group of secretory cell types that differ in their developmental origin, some even arising from different germ layers (highlighted edges in *Figure 2B*). Within ectoderm, frog cement gland cells map to zebrafish $muc5ac^+$ secretory epidermal cells, and frog small secretory cells (SSCs) map to zebrafish $pvalb8^+$ mucous cells (*Janicke et al., 2010*). Across germ layers, SSCs also map weakly to zebrafish endodermal cells, and frog ectodermal hatching gland maps to zebrafish mesodermal hatching gland. These cell types are linked through a large set of genes, including proteins involved in vesicular protein trafficking and several conserved transcription factors (TFs) such as *myb*, *foxa1*, *xbp1*, and *klf17* (*Figure 2D*), which all have documented functions in controlling the differentiation of secretory cell types (*Bennett et al., 2007*; *Dubaissi et al., 2014*; *Pan et al., 2014*). For example, *klf17* is expressed in zebrafish and frog hatching glands, and plays essential roles in regulating gland cell specification in both species (*Kurauchi et al., 2010*; *Suzuki et al., 2019*). Together, the conserved cell type specification programs (*Erwin and Davidson, 2009*) between developmentally distinct secretory cells support the notion that they may be transcriptionally and evolutionarily related despite having different developmental origins (*Arendt et al., 2016*).

To benchmark the performance of SAMap, we used eggNOG (*Huerta-Cepas et al., 2019*) to define one-to-one vertebrate orthologs between zebrafish and frog and fed these gene pairs as input to several broadly used single-cell data integration methods, Seurat (*Stuart et al., 2019*), LIGER (*Welch et al., 2019*), Harmony (*Korsunsky et al., 2019*), Scanorama (*Hie et al., 2019*), and BBKNN (*Polański et al., 2019*). We found that they failed to map the two atlases, yielding minimal alignment between them (*Figure 2E* and *Figure 2—figure supplement 1*). We also compared the results when restricting SAMap to using the one-to-one orthologs instead of the full homology graph. Even when removing the many-to-many gene homologies and the iterative refinement of the

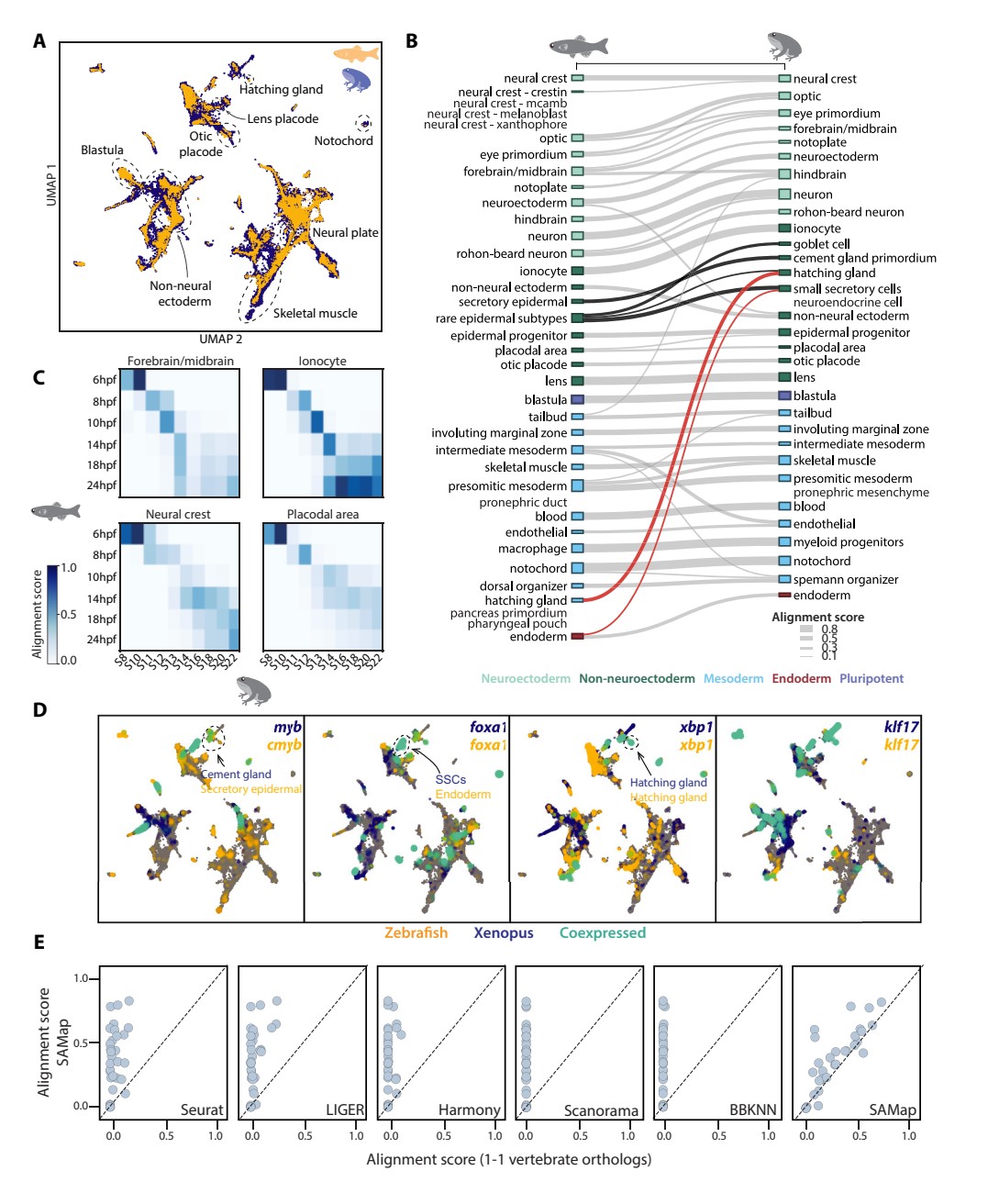

**Figure 2.** SAMap successfully maps *D. rerio* and *X. tropicalis* atlases. (**A**) UMAP projection of the combined zebrafish (yellow) and *Xenopus* (blue) manifolds, with example cell types circled. (**B**) Sankey plot summarizing the cell type mappings. Edges with alignment score <0.1 are omitted. Edges that connect developmentally distinct secretory cell types are highlighted in black, with connections across germ layers highlighted in red. (**C**) Heatmaps of alignment scores between developmental time points for ionocyte, forebrain/midbrain, placodal, and neural crest lineages. X-axis: *Xenopus*. Y-axis: zebrafish. (**D**) Expressions of orthologous gene pairs linked by SAMap are overlaid on the combined UMAP projection. Expressing cells are color-coded by species, with those connected across species colored cyan. Cells with no expression are shown in gray. The mapped secretory cell types are highlighted with circles. (**E**) SAMap alignment scores compared to those of benchmarking methods using one-to-one vertebrate orthologs as input. Each dot represents a cell type pair supported by ontogeny annotations.

The online version of this article includes the following figure supplement(s) for figure 2:

**Figure supplement 1.** Existing methods failed to map *D. rerio* and *X. tropicalis* atlases.

homology graph, we identified similar, albeit weaker, cell type mappings. This suggests that, at least for the frog and zebrafish comparison, SAMap's performance is owed in large part to its robust, atlas stitching approach.

Finally, to test if SAMap is robust to incomplete cell type atlases, we downsampled the frog and zebrafish data by systematically removing individual cell types. We found that cell types whose homologous partners were removed mapped weakly to closely related cell types, some of which were already present in the original mapping. For example, optic cells from both species were also connected to eye primordium, frog skeletal muscles to zebrafish presomitic mesoderm, and frog hindbrain to zebrafish forebrain/midbrain (*Supplementary file 3*). We observed several new mapping pairs, but their alignment scores were all barely above the detection threshold of SAMap. Moreover, most of these edges were mapped between cell types with similar developmental origins, with the only exception being the zebrafish neural crest mapped to the frog otic placode in the absence of frog neural crest cells. Examining the genes that support this mapping revealed that both cell types express *sox9* and *sox10*, two TFs previously implicated to form a conserved gene regulatory circuit common to otic/neural crest cells (*Betancur et al., 2011*). Taken together, these results suggest that SAMap is more sensitive in linking homologous cell types compared to other existing methods, exhibits high robustness when applied to incomplete datasets, and yields mapping results that are well supported by conserved gene expression programs.

## Paralog substitutions are prevalent between homologous cell types in frog and zebrafish

The key benefit of using the full homology graph is to enable the systematic identification of gene paralogs that exhibit greater similarity in expression across species than their corresponding orthologs. These events are expected to arise as the result of gene duplications followed by diversification of the resulting in-paralogs (*Studer and Robinson-Rechavi, 2009*). In an alternative scenario, genetic compensation by transcriptional adaptation, where loss-of-function mutations are balanced by upregulation of related genes with similar sequences (*El-Brolosy et al., 2019*), could also result in this signature.

In total, SAMap selected 8286 vertebrate orthologs and 7093 paralogs, as enumerated by eggNOG, for manifold alignment. Paralogs were identified as non-orthologous genes that map to the same eggNOG orthology group ancestral to Vertebrata. Among these, 565 genes have markedly higher expression correlations (correlation difference >0.3) with their paralogs than their orthologs (see *Figure 3A* for examples), and 209 of them have orthologs that are either completely absent or lowly expressed with no cell-type specificity (*Supplementary file 4*). We term these events as 'paralog substitutions', as the orthologs may have lost or changed their functional roles at some point and were compensated for by their paralogs. Substituting paralogs were identified in most cells types with some (e.g. dorsal organizer) exhibiting higher rates than others (*Figure 3B*), suggesting uneven diversification rates of paralogs across cell types. SAMap also linked an additional 297 homologous pairs previously unannotated by orthology or paralogy, but which exhibit sequence similarity and high expression correlations (>0.5 Pearson correlation). These likely represent unannotated orthologs/paralogs or isofunctional, distantly related homologs (*Gabaldón and Koonin, 2013*).

We next asked whether paralog substitution rates depend on the evolutionary time since gene duplication. We categorized paralogs by the taxonomic level of their most recent shared orthology group and found that more recent paralogs substitute orthologs at higher rates than more ancient paralogs (*Figure 3C*). This observation is consistent with the expectation that less diverged genes may be more capable of functionally compensating for each other. To rule out the possibility that these paralogs were linked spuriously during the homology refinement steps of SAMap, we repeated the paralog substitution analysis on an aligned manifold constructed using only one-to-one orthologs. We identified 70% of the paralog substitutions and observed similar patterns in evolutionary time and cell type dependencies (*Figure 3—figure supplement 1A–B*). The other 30% of substitutions had smaller correlation differences on the border of our detection threshold (i.e. correlation difference >0.3) (*Figure 3—figure supplement 1C*). Failure to detect these substitutions was due to inaccurate imputation of gene expressions across species when restricting the mapping to one-to-one orthologs, which resulted in weaker alignment with fewer cross-species edges. Altogether, these results illustrate the potential of SAMap in leveraging single-cell gene expression data for pruning

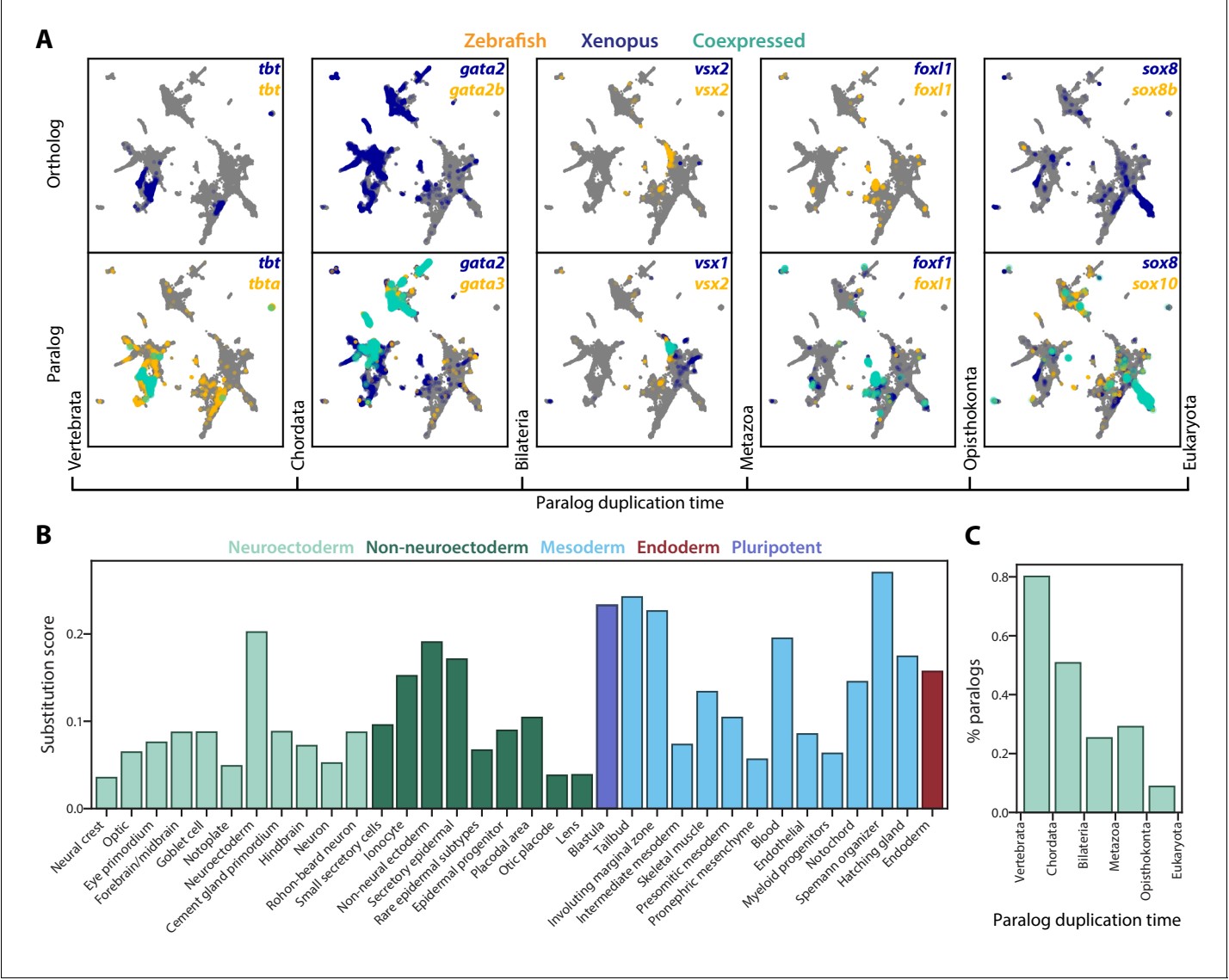

**Figure 3.** SAMap reveals prevalent paralog substitutions in frog and zebrafish. (**A**) Expression of orthologous (top) and paralogous (bottom) gene pairs overlaid on the combined UMAP projection. Expressing cells are color-coded by species, with those that are connected across species colored cyan. Cells with no expression are shown in gray. Paralogs are ordered by the evolutionary time when they are inferred to have duplicated. (**B**) Paralog substitution scores of all cell types. The substitution score counts the number of substituting paralogs that are differentially expressed in a particular cell type while normalizing for the number of differentially expressed genes in a cell type and the number of paralogs of a gene (see Materials and methods). (**C**) The percentage of paralogs from each phylogenetic age that were substituted for orthologs in frog or zebrafish lineages.

The online version of this article includes the following figure supplement(s) for figure 3:

**Figure supplement 1.** Paralog substitution analysis yields similar results using the SAMap manifold constructed from one-to-one orthologs.

the networks of homologous genes to identify evolutionary substitution of paralogs and, more generally, identify non-orthologous gene pairs that may perform similar functions in the cell types within which they are expressed.

## Homologous cell types between two flatworm species with divergent body plans

To test if we can identify homologous cell types in animals with radically different body plans, we mapped the cell atlases of two flatworms, the planarian *Schmidtea mediterranea* (*Fincher et al.,*

2018), and the trematode *Schistosoma mansoni*, which we collected recently (*Li et al., 2021*). They represent two distant lineages within the same phylum but have remarkably distinct body plans and autecology (*Laumer et al., 2015*; *Littlewood and Waeschenbach, 2015*). While planarians live in freshwater and are known for their ability to regenerate (*Reddien, 2018*), schistosomes live as parasites in humans. The degree to which cell types are conserved between them is unresolved, given the vast phenotypic differences caused by the transition from free-living to parasitic habits (*Laumer et al., 2015*).

SAMap revealed broad cell type homology between schistosomes and planarians. The schistosome had cells mapped to the planarian stem cells, called neoblasts, as well as most of the differentiated tissues: neural, muscle, intestine, epidermis, parenchymal, protonephridia, and *cathepsin*⁺ cells, the latter of which consists of cryptic cell types that, until now, have only been found in planarians (*Fincher et al., 2018*; *Figure 4A*). These mappings are supported by both known cell-type-

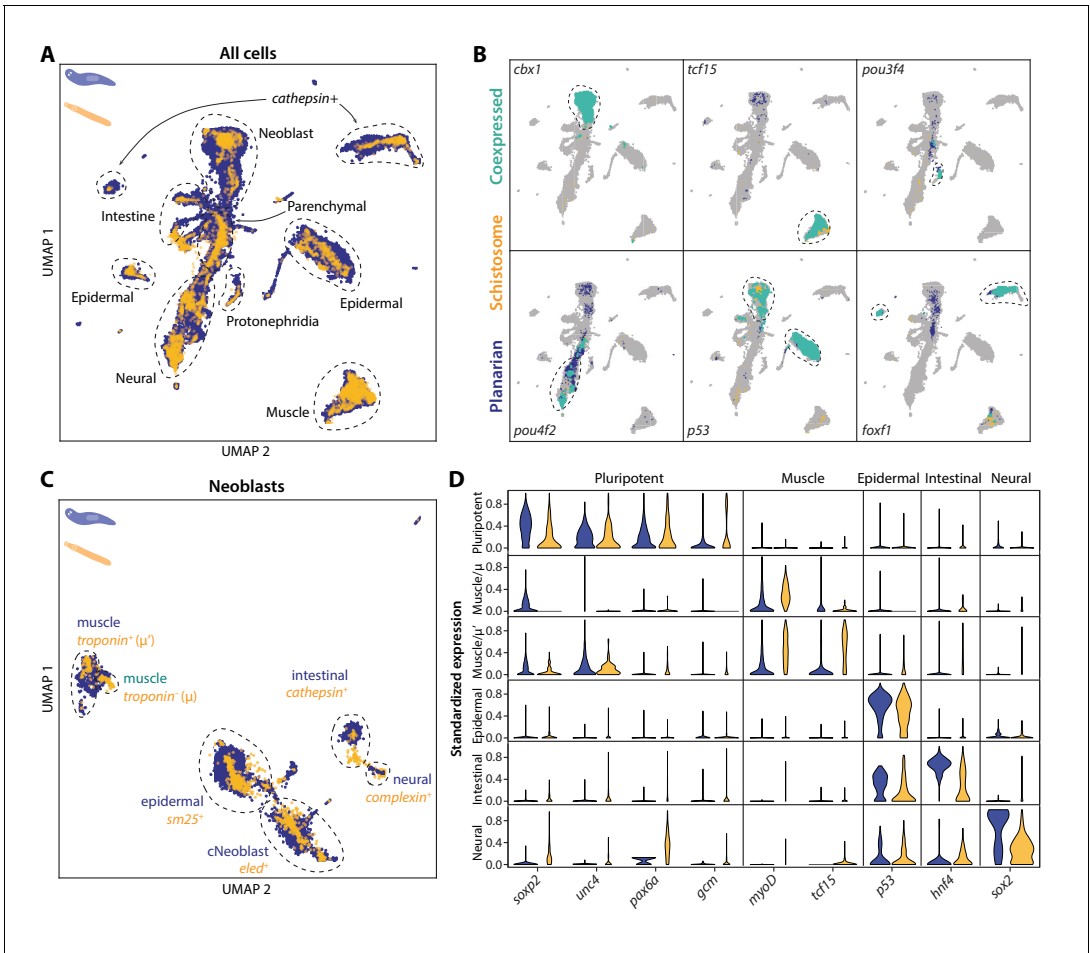

**Figure 4.** SAMap transfers cell type information from a well-annotated organism (planarian *S. mediterranea*) to its less-studied cousin (schistosome *S. mansoni*) and identifies parallel stem cell compartments. (**A**) UMAP projection of the combined manifolds. Tissue type annotations are adopted from the *S. mediterranea* atlas (*Fincher et al., 2018*). The schistosome atlas was collected from juvenile worms, which we found to contain neoblasts with an abundance comparable to that of planarian neoblasts (*Li et al., 2021*). (**B**) Overlapping expressions of selected tissue-specific TFs with expressing cell types circled. (**C**) UMAP projection of the aligned manifolds showing planarian and schistosome stem cells, with homologous subpopulations circled. Planarian neoblast data is from *Zeng et al., 2018*, and cNeoblasts correspond to the Nb2 population, which are pluripotent cells that can rescue neoblast-depleted planarians in transplantation experiments. (**D**) Distributions of conserved TF expressions in each neoblast subpopulation. Expression values are *k*-nearest-neighbor averaged and standardized, with negative values set to zero. Blue: planarian; yellow: schistosome.

The online version of this article includes the following figure supplement(s) for figure 4:

**Figure supplement 1.** SAMap-linked gene pairs that are enriched in cell type pairs between *S. mediterranea* and *S. mansoni*.

**Figure supplement 2.** Schistosome muscle progenitors express canonical muscle markers.

specific marker genes and numerous homologous transcriptional regulators (*Figure 4B* and *Figure 4—figure supplement 1*).

We next determined if cell type homologies exist at the subtype level. For this, we compared the stem cells, as planarian neoblasts are known to comprise populations of pluripotent cells and tissue-specific progenitors (*Fincher et al., 2018*; *Zeng et al., 2018*). By mapping the schistosome stem cells to a planarian neoblast atlas (*Zeng et al., 2018*), we found that the schistosome has a population of stem cells, ε-cells (*Wang et al., 2018*), that cluster with the planarian's pluripotent neoblasts, both expressing a common set of TFs (e.g. *soxp2, unc4, pax6a, gcm1*) (*Figure 4C–D*). The ε-cells are closely associated with juvenile development and lost in adult schistosomes (*Wang et al., 2018*; *Nanes Sarfati et al., 2021*), indicating pluripotent stem cells may be a transient population restricted to their early developmental stages. This is consistent with the fact that, whereas schistosomes can heal wounds, they have limited regenerative ability (*Wendt and Collins, 2016*). SAMap also linked other schistosome stem cell populations with planarian progenitors, including two populations of schistosome stem cells – denoted as μ (*Tarashansky et al., 2019*) and μ' – to planarian muscle progenitors, all of which express *myoD*, a canonical master regulator of myogenesis (*Scimone et al., 2017*). These likely represent early and late muscle progenitors, respectively, as μ-cells do not yet express differentiated muscle markers such as *troponin*, whereas μ'-cells do (*Figure 4—figure supplement 2*).

## Cell type families spanning the animal tree of life

To compare cell types across broader taxonomic scales, we extended our analysis to include juvenile freshwater sponge (*Spongilla lacustris*) (*Musser et al., 2019*), adult *Hydra* (*Hydra vulgaris*) (*Siebert et al., 2019*), and mouse (*Mus musculus*) embryogenesis (*Pijuan-Sala et al., 2019*) atlases. In total, SAMap linked 1051 cross-species pairs of cell types, defined by the annotations used in each respective study. Of the cell type pairs, 95% are supported by at least 40 enriched gene pairs, and 87% are supported by more than 100 gene pairs, indicating that SAMap does not spuriously connect cell types with limited overlap in transcriptional profiles (*Figure 5—figure supplement 1A*).

We next extended the notion of cell type pairs to cell type trios, as mapped cell types gain additional support if they share transitive relationships to other cell types through independent mappings, forming cell type triangles among species. The transitivity of a cell type pair (edge) or a cell type (node) can be quantified as the fraction of triads to which they belong that form triangles (*Figure 5A*). The majority (81%) of cell type pairs have non-zero transitivity independent of alignment score and the number of enriched gene pairs (*Figure 5—figure supplements 1–2*). Cell type pairs with fewer than 40 enriched gene pairs tend to have lower (<0.4) transitivity (*Figure 5—figure supplement 1B*). The transitivity measure can also be used to identify potentially spurious connections. 16% of mapped cell type pairs have zero edge transitivity but non-zero node transitivity. These cell types are connected to only a single member of an interconnected cell type group (motifs 2 and 3 in *Figure 5B*). Such links may be of lower confidence as they should connect to other members of the group and are thus excluded from downstream analysis.

Among the interconnected groups of cell types, we identified families of neural cells and contractile cells (*Figure 5C*). Both cell type families are highly transitive compared to the overall graph transitivity (bootstrap p-value<$1\times10^{-5}$), meaning that their constituent cell types have more transitive edges within the group than outside the group (*Figure 5D*). In addition, the dense, many-to-many connections within the contractile and neural families are each supported by at least 40 enriched gene pairs (*Figure 5E*). Consistent with the nerve net hypothesis suggesting a unified origin of neural cell types (*Tosches and Arendt, 2013*), the neural family includes vertebrate brain tissues, both bilaterian and cnidarian neurons, cnidarian nematocytes that share the excitatory characteristics of neurons (*Weir et al., 2020*), and *Spongilla* choanocytes and apopylar cells, both of which are not considered as neurons but have been shown to express postsynaptic-like scaffolding machinery (*Musser et al., 2019*; *Wong et al., 2019*). The contractile family includes myocytes in bilaterian animals, *Hydra* myoepithelial cells that are known to have contractile myofibrils (*Buzgariu et al., 2015*), and sponge pinacocytes and myopeptidocytes, both of which have been implicated to play roles in contractility (*Musser et al., 2019*; *Sebé-Pedrós et al., 2018*). In contrast to the families encompassing all seven species, we also found a fully interconnected group that contains invertebrate multipotent stem cells, including planarian and schistosome neoblasts, *Hydra* interstitial cells, and sponge archeocytes (*Alié et al., 2015*). The lack of one-to-one connections across phyla is in keeping with

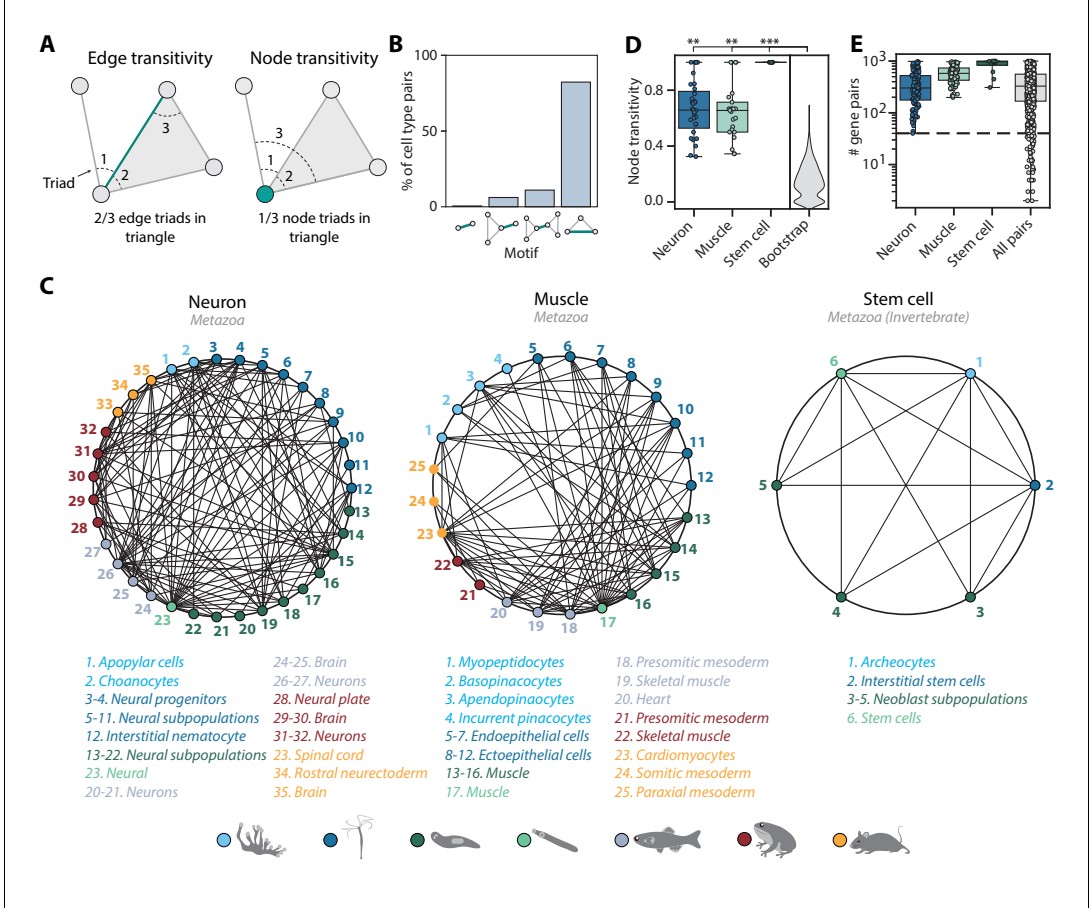

**Figure 5.** Mapping evolutionarily distant species identifies densely connected cell type groups. (A) Schematic illustrating edge (left) and node (right) transitivities, defined as the fraction of triads (set of three connected nodes) in closed triangles. (B) The percentage of cell type pairs that are topologically equivalent to the green edge in each illustrated motif. (C) Network graphs showing highly connected cell type families. Each node represents a cell type, color-coded by species (detailed annotations are provided in ***Supplementary file 7***). Mapped cell types are connected with an edge. (D) Boxplot showing the median and interquartile ranges of node transitivities for highly connected cell type groups. For all box plots, the whiskers denote the maximum and minimum observations. The average node transitivity per group is compared to a bootstrapped null transitivity distribution, generated by repeatedly sampling subsets of nodes in the cell type graph and calculating their transitivities. **$p < 5 \times 10^{-5}$, ***$p < 5 \times 10^{-7}$. (E) Boxplot showing the median and interquartile ranges of the number of enriched gene pairs in highly connected cell type groups. All cell type connections in these groups have at least 40 enriched gene pairs (dashed line).

The online version of this article includes the following figure supplement(s) for figure 5:

**Figure supplement 1.** Number of enriched gene pairs are mostly independent of edge transitivity.

**Figure supplement 2.** Alignment scores are mostly independent of edge transitivity.

recent hypotheses that ancestral cell types diversified into families of cell types after speciation events (*Arendt et al., 2016*; *Arendt et al., 2019*). Our findings thus suggest that these cell type families diversified early in animal evolution.

## Transcriptomic signatures of cell type families

The high interconnectedness between cell types across broad taxonomic scales suggests that they should share ancestral transcriptional programs (*Arendt et al., 2016*). SAMap identified broad transcriptomic similarity between bilaterian and non-bilaterian contractile cells that extends beyond the core contractile apparatus. It links a total of 23601 gene pairs, connecting 5471 unique genes, which are enriched in at least one contractile cell type pair. Performing functional enrichment analysis on these genes, we found cytoskeleton and signal transduction functions to be enriched (p-value<$10^{-3}$) based on the KOG functional classifications (*Tatusov et al., 2003*) assigned by eggNOG

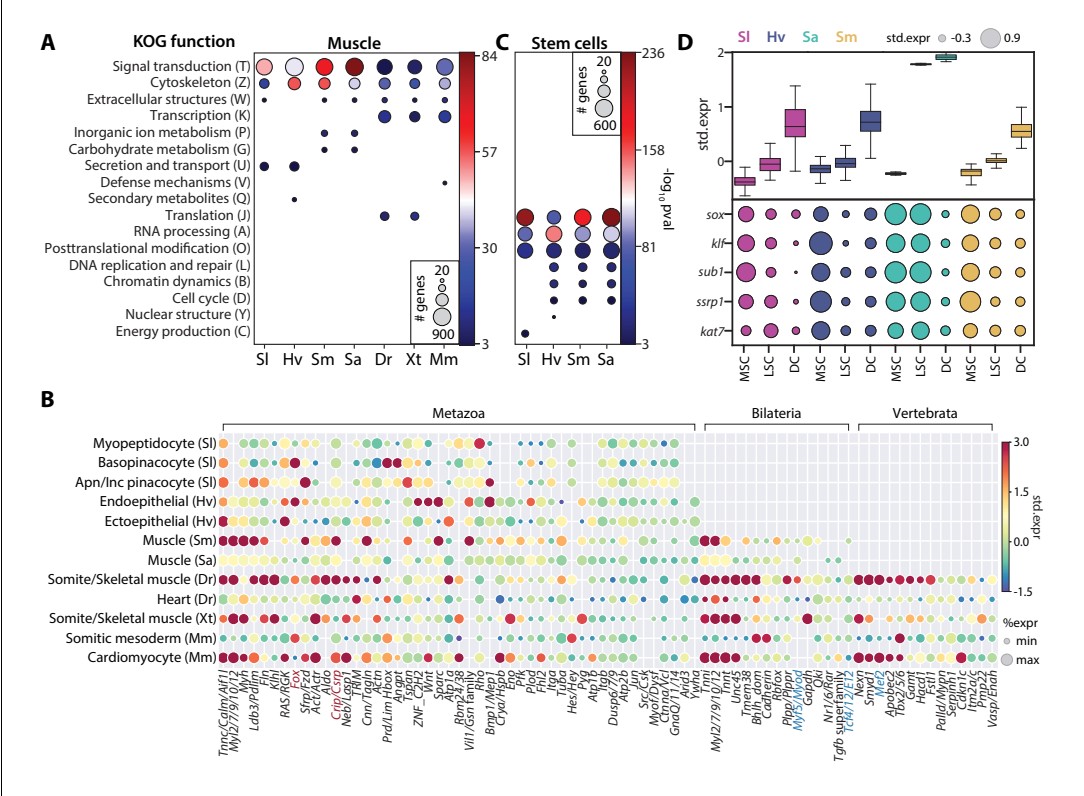

**Figure 6.** SAMap identifies muscle and stem cell transcriptional signatures conserved across species. (**A**) Enrichment of KOG functional annotations calculated for genes shared in contractile cell types. For each species, genes enriched in individual contractile cell types are combined. (**B**) Expression and enrichment of conserved muscle genes in contractile cell types. Color: mean standardized expression. Symbol size: the fraction of cells each gene is expressed in per cell type. Homologs are grouped based on overlapping eukaryotic eggNOG orthology groups. If multiple genes from a species are contained within an orthology group, the gene with highest standardized expression is shown. Genes in blue: core transcriptional program of bilaterian muscles; red: transcriptional regulators conserved throughout Metazoa. (**C**) Enrichment of KOG functional annotations for genes shared by stem cell types. (**D**) Top: boxplot showing the median and interquartile ranges of the mean standardized expressions of stem cell-enriched genes in multipotent stem cells (MSCs), lineage-committed stem cells (LSCs), and differentiated cells (DCs). MSCs include sponge archaeocytes (*Musser et al., 2019*), hydra interstitial stem cells (*Siebert et al., 2019*), planarian neoblasts cluster 0 defined in *Fincher et al., 2018*, schistosome ε-cells (*Tarashansky et al., 2019*). LSCs include sponge transition cells, hydra ecto- and endo-epithelial stem cells; planarian *piwi+* cells that cluster with differentiated tissues, and schistosome tissue-specific progenitors. Bottom: dot plot showing the mean standardized expressions of selected transcriptional regulators. The transcript IDs corresponding to each gene are listed in *Supplementary file 6*.

The online version of this article includes the following figure supplement(s) for figure 6:

**Figure supplement 1.** Phylogenetic reconstruction of animal contractile cell transcriptional regulators.

(*Figure 6A*). These genes include orthology groups spanning diverse functional roles in contractile cells, including members of the adhesion complex that connects cells, actomyosin networks that drive contractility, and signaling pathways that stimulate contraction (*Figure 6B* and *Supplementary file 5*). This observation suggests that contractile cells near the beginning of animal evolution already possessed the broad assemblage of gene modules associated with different functional aspects of derived muscle cell types in extant animals.

We also identified several transcriptional regulators shared among contractile cells (*Figure 6B*). Previously known core regulators involved in myocyte specification (*Brunet et al., 2016*) were enriched only in bilaterian (e.g. *myod* and *tcf4/E12*) or vertebrate contractile cells (e.g. *mef2*). In contrast, we found homologs of Muscle Lim Protein (*Csrp*) and Forkhead Box Group 1 (*Larroux et al., 2008*) enriched in contractile cells from all seven species. The Fox proteins included FoxC, which is known to regulate cardiac muscle identity in vertebrates (*Brunet et al., 2016*) and is contractile-specific in all species except schistosome and *Spongilla*. Notably, we also identified FoxG orthologs to

be enriched in three of the four invertebrates (*Figure 6—figure supplement 1*), suggesting that FoxG may play an underappreciated role in contractile cell specification outside vertebrates. Whether the most conserved regulators are positioned at the center of the cell type specification regulatory complex remains an important question to address in future studies.

For the family of invertebrate multipotent stem cells, we identified 3343 genes that are enriched in at least one cell type pair and observed significant enrichment (p-value<$10^{-3}$) of genes involved in translational regulation such as RNA processing, translation, and post-translational modification (*Figure 6C*). We also observed cell cycle and DNA replication genes, as expected for actively dividing cells, though these were not the most enriched categories. Shared stem cell genes comprise 979 orthology groups, 17% of which are enriched in all cell types of this family (*Supplementary file 5*). Importantly, these genes have consistently higher expression in the multipotent stem cells (MSCs) compared to lineage-restricted stem cells (LSCs) and differentiated cells (DCs) for all four species (*Figure 6D*), and may represent a large, deeply conserved gene module specifically associated with multipotency.

Next, we asked whether there is an MSC-specific transcriptional regulatory program. Notably, we identified a number of transcriptional regulators enriched in MSCs across all four invertebrates. This included TFs from *sox* and *klf* orthology groups, the transcriptional coactivator *sub1*, and several chromatin modifiers, including *ssrp1*, a subunit of the FACT complex, and *kat7*, a histone acetyltransferase (*Figure 6D* and *Supplementary file 6*). *sox* and *klf* are well-known pluripotency factors (*Bialkowska et al., 2017*; *Sarkar and Hochedlinger, 2013*), whereas the others have been studied in various processes associated with stem cell functions (*MacPherson et al., 2020*; *Sikder et al., 2019*; *Zeng et al., 2013*). The conserved enrichment of these transcriptional regulators in invertebrate MSCs suggests that their functional role in stem cells may be more phylogenetically ancient than previously appreciated. Determining their roles in establishing and maintaining multipotency across diverse animal taxa is an important avenue of future research.

## Discussion

Cell types evolve as their gene expression programs change either as integrated units or via evolutionary splitting that results in separate derived programs. While this notion of coupled cellular and molecular evolution has gained significant traction in the past years, systematically comparing cell type-specific gene expression programs across species has remained a challenging problem. Here, we map single-cell atlases between evolutionarily distant species in a manner that accounts for the complexity of gene evolution. SAMap aligns cell atlases in two mutually reinforcing directions, mapping both the genes and the cells, with each feeding back into the other. Although this algorithm scales well to the majority of presently available whole-organism cell atlases, the size of single-cell datasets will continue to increase. The most memory-intensive steps in SAMap are the neighborhood coarsening of cross-species edges and the cross-species imputation of gene expression to calculate gene-gene correlations, both of which could be intractable for datasets containing millions of cells. Our current solution is to chunk these operations into smaller blocks for large datasets to avoid memory limitations, but the runtime increases significantly as a result. An option for mapping massive datasets may be to downsample each atlas. So long as all cell types are preserved and remain separable during downsampling, we expect the mapping results to be the same.

SAMap allows us to identify one-to-one cell type concordance between animals in the same phylum, whereas between phyla, we observe interconnected cell types forming distinct families. These findings support the notion that cell types evolve via hierarchical diversification (*Arendt et al., 2019*), resulting in cell type families composed of evolutionarily related cell types sharing a regulatory gene expression program that originated in their common ancestor. One-to-one cell type homologies should exist only if no further cell type diversification has occurred since the speciation. To understand the genetic underpinnings of the observed cell type homologies, we have examined similarities in gene expression programs of several cell types in depth. Among various gene sets, we have focused on TFs, as they form regulatory networks that specify cell type identities and activate downstream differentiation gene batteries (*Erwin and Davidson, 2009*). Notably, many of the aligned cell types we identified share expression of transcription factors known to play important roles in cell type differentiation, suggesting SAMap alignments based on mutual connectivity reflect evolutionary homology, rather than convergent functional similarity. SAMap thus provides a

roadmap for tracing cell type evolutionary history and identifying the molecular changes in transcription factor regulatory complexes that have driven cell type diversification.

In parallel, SAMap systematically identifies instances where paralogs exhibit greater expression similarity than orthologs across species. Paralog substitution can occur due to differential loss or retention of cell-type-specific expression patterns of genes that were duplicated in the common ancestor (*Shafer et al., 2020*; *Studer and Robinson-Rechavi, 2009*) or due to compensating upregulation of paralogs following a loss-of-function mutation acquired by an ortholog (*El-Brolosy et al., 2019*). Considering our observation that paralog substitutions occur at higher rates for more recent paralogs, which should be more capable of functionally compensating for each other, we expect the latter scenario to be more likely, at least between frog and zebrafish. Paralog substitutions may also play an important role in cell type diversification, enabling newly evolved sister cell types to subfunctionalize via the use of distinct paralogs. Whereas the analysis presented here focuses on comparisons between two species, incorporating multiple species into a single analysis that also accounts for their phylogenetic relatedness could enable determining the stability of paralog substitutions within clades and their associated cell type diversification events. However, this will require datasets that densely sample species within specific clades and at key branching points along the tree of life.

Besides applications in evolutionary biology, we anticipate SAMap can catalyze the annotation of new cell atlases from non-model organisms, which often represents a substantial bottleneck requiring extensive manual curation and prior knowledge. Its ability to use the existing atlases to inform the annotation of cell types in related species will keep improving as more datasets become available to better sample the diversity of cell types throughout the animal kingdom.

# Materials and methods

## Data and code availability

The source code for SAMap is publicly available at Github (https://github.com/atarashansky/SAMap; copy archived at swh:1:rev:c696585f8fe41ec1599b0720df579f3cb14f935b; *Tarashansky et al., 2021*), along with the code to perform the analysis and generate the types of plots presented in the figures. We also provide a wrapper function to launch a graphical user interface provided by the SAM package to interactively explore both datasets in the combined manifold. The datasets analyzed in this study are detailed in *Supplementary file 1* with their accessions and annotations provided.

## The SAMap algorithm

The SAMap algorithm contains three major steps: preprocessing, mutual nearest neighborhood alignment, and gene-gene correlation initialization. The latter two are repeated for three iterations, by default, to balance alignment performance and computational runtime.

### Preprocessing
#### Generate gene homology graph via reciprocal BLAST

We first construct a gene-gene bipartite graph between two species by performing reciprocal BLAST of their respective transcriptomes using *tblastx*, or proteomes using *blastp. tblastn* and *blastx* are used for BLAST between proteome and transcriptome. When a pair of genes share multiple High Scoring Pairs (HSPs), which are local regions of matching sequences, we use the HSP with the highest bit score to measure homology. Only pairs with E-value $<10^{-6}$ are included in the graph.

Although we define similarity using BLAST, SAMap is compatible with other protein homology detection methods (e.g. HMMER [*Eddy, 2008*]) or orthology inference tools (e.g. OrthoClust [*Yan et al., 2014*] and eggNOG [*Huerta-Cepas et al., 2019*]). While each of these methods has known strengths and limitations, BLAST is chosen for its broad usage, technical convenience, and compatibility with low-quality transcriptomes.

We encode the BLAST results into two triangular adjacency matrices, $A$ and $B$, each containing bit scores in one BLAST direction. We combine $A$ and $B$ to form a gene-gene adjacency matrix $G$. After symmetrizing $G$, we remove edges that only appear in one direction: $G = Recip\left(\frac{1}{2}\left[(A+B) + (A+B)^T\right]\right) \in \Re^{m_1+m_2 \times m_1+m_2}$, where $Recip$ only keeps reciprocal edges, and $m_1$

and $m_2$ are the number of genes of the two species, respectively. To filter out relatively weak homologies, we also remove edges where $G_{ab} < 0.25 \max_b(G_{ab})$. Edge weights are then normalized by the maximum edge weight for each gene and transformed by a hyperbolic tangent function to increase discriminatory power between low and high edge weights,

$$G_{ab} = 0.5 + 0.5 tanh\left(10 G_{ab} / \max_b(G_{ab}) - 5\right).$$

## Construct manifolds for each cell atlas separately using the SAM algorithm

The single-cell RNAseq datasets are normalized such that each cell has a total number of raw counts equal to the median size of single-cell libraries. Gene expressions are then log-normalized with the addition of a pseudocount of 1. Genes expressed (i.e. $log_2(D + 1) > 1$) in greater than 96% of cells are filtered out. SAM is run using the following parameters: *preprocessing = 'StandardScaler'*, *weight_PCs = False*, *k = 20*, and *npcs = 150*. A detailed description of parameters is provided previously (*Tarashansky et al., 2019*). SAM outputs $N_1$ and $N_2$, which are directed adjacency matrices that encode *k*-nearest neighbor graphs for the two datasets, respectively.

SAM only includes the top 3000 genes ranked by SAM weights and the first 150 principal components (PCs) in the default mode to reduce computational complexity. However, downstream mapping requires PC loadings for all genes. Thus, in the final iteration of SAM, we run PCA on all genes and take the top 300 PCs. This step generates a loading matrix for each species $i$, $L_i \in \Re^{300 \times m_i}$.

## Mutual nearest neighborhood alignment
### Transform feature spaces between species

For the gene expression matrices $Z_i \in \Re^{n_i \times m_i}$, where $n$ and $m$ are the number of cells and genes respectively, we first zero the expression of genes that do not have an edge in $G$ and standardize the expression matrices such that each gene has zero mean and unit variance, yielding $\tilde{Z}_i$. $G$ represents a bipartite graph in the form of $G = \begin{bmatrix} 0_{m_1,m_1} & H \in \Re^{m_1 \times m_2} \\ H^T \in \Re^{m_2 \times m_1} & 0_{m_2,m_2} \end{bmatrix}$, where $0_{m,m}$ is $m \times m$ zero matrix and $H$ is the biadjacency matrix. Letting $H_1 = H$ and $H_2 = H^T$ encoding directed edges from species 1 to 2 and 2 to 1, respectively, we normalize the biadjacency matrix $H_i$ such that each row sums to 1: $H_i = SumNorm(H_i) \in \Re^{m_i \times m_j}$, where the *SumNorm* function normalizes the rows to sum to 1. The feature spaces can be transformed between the two species via weighted averaging of gene expression, $\tilde{Z}_{ij} = \tilde{Z}_i H_i$.

### Project single-cell gene expressions into a joint PC space

We project the expression data from two species into a joint PC space (*Barkas et al., 2019*), $P_i = \tilde{Z}_i L_i^T$ and $P_{ij} = \tilde{Z}_{ij} L_j^T$. We then horizontally concatenate the principal components $P_i$ and $P_{ij}$ to form $P_i \in \Re^{n_i \times 600}$.

### Calculate k-nearest cross-species neighbors for all cells

Using the joint PCs, $P_i$, we identify for each cell the *k*-nearest neighbors in the other dataset using cosine similarity ($k = 20$ by default). Neighbors are identified using the *hnswlib* library, a fast approximate nearest-neighbor search algorithm (*Malkov and Yashunin, 2020*). This outputs two directed biadjacency matrices $C_i \in \Re^{n_i \times n_j}$ for $(i,j) = (1,2)$ or $(2,1)$ with edge weights equal to the cosine similarity between the PCs.

### Apply the graph-coarsening mapping kernel to identify cross-species mutual nearest neighborhoods

To increase the stringency and confidence of mapping, we only rely on cells that are *mutual* nearest cross-species neighbors, which are typically defined as two cells reciprocally connected to one another (*Haghverdi et al., 2018*). However, due to the noise in cell-cell correlations and stochasticity in the kNN algorithms, cross-species neighbors are often randomly assigned from a pool of cells that appear equally similar, decreasing the likelihood of mutual connectivity between individual cells even if they have similar expression profiles. To overcome this limitation, we integrate information

from each cell's local neighborhood to establish more robust mutual connectivity between cells across species. Two cells are thus defined as mutual nearest cross-species neighbors when their respective neighborhoods have mutual connectivity.

Specifically, the nearest neighbor graphs $N_i$ generated by SAM are used to calculate the neighbors of cells $t_i$ hops away along outgoing edges: $\bar{N}_i = N_i^{t_i}$, where $\bar{N}_i$ are adjacency matrices that contain the number of paths connecting two cells $t_i$ hops away, for $i = 1$ or 2. $t_i$ determines the length-scale over which we integrate incoming edges for species $i$. Its default value is 2 if the dataset size is less than 20,000 cells and 3 otherwise. However, cells within tight clusters may have spurious edges connecting to other parts of the manifold only a few hops away. To avoid integrating neighborhood information outside this local structure, we use the Leiden algorithm (*Traag et al., 2019*) to cluster the graph and identify a local neighborhood size for each cell (the resolution parameter is set to 3 by default). If cell $a$ belongs to cluster $c_a$, then its neighborhood size is $l_a = |c_a|$. For each row $a$ in $\bar{N}_i$ we only keep the $l_a$ geodesically closest cells, letting the pruned graph update $N_i$.

Edges outgoing from cell $a_i$ in species $i$ are encoded in the corresponding row in the adjacency matrix: $C_{i,a_i}$. We compute the fraction of the outgoing edges from each cell that target the local neighborhood of a cell in the other species: $\tilde{C}_{i,a_i b_j} = \sum_{c \in X_{j,b_j}} C_{i,a_i c}$, where $X_{j,b_j}$ is the set of cells in the neighborhood of cell $b_j$ in species $j$ and $\tilde{C}_{i,a_i b_j}$ is the fraction of outgoing edges from cell $a_i$ in species $i$ targeting the neighborhood of cell $b_j$ in species $j$.

To reduce the density of $\tilde{C}_i$ so as to satisfy computational memory constraints, we remove edges with weight less than 0.1. Finally, we apply the mutual nearest neighborhood criterion by taking the element-wise, geometric mean of the two directed bipartite graphs: $\tilde{C} = \sqrt{\tilde{C}_1 \circ \tilde{C}_2}$. This operation ensures that only bidirectional edges are preserved, as small edge weights in either direction results in small geometric means.

## Assign the k-nearest cross-species neighborhoods for each cell

Given the mutual nearest neighborhoods $\tilde{C} \in \Re^{n_1 \times n_2}$, we select the $k$ nearest neighborhoods for each cell in both directions to update the directed biadjacency matrices $C_1$ and $C_2$: $C_1 = KNN\left(\tilde{C}, k\right)$ and $C_2 = KNN\left(\tilde{C}^T, k\right)$, with $k = 20$ by default.

## Stitch the manifolds

We use $C_1$ and $C_2$ to combine the manifolds $N_1$ and $N_2$ into a unified graph. We first weight the edges in $N_1$ and $N_2$ to account for the number of shared cross-species neighbors by computing the one-mode projections of $C_1$ and $C_2$. In addition, for cells with strong cross-species alignment, we attenuate the weight of their within-species edges. For cells with little to no cross-species alignment, their within-species are kept the same to ensure that the local topological information around cells with no alignment is preserved.

Specifically, we use $N_1$ and $N_2$ to mask the edges in the one-mode projections, $\tilde{N}_1 = U(N_1) \circ (Norm(C_1)Norm(C_2))$ and $\tilde{N}_2 = U(N_2) \circ (Norm(C_2)Norm(C_1))$, where $U(E)$ sets all edge weights in graph $E$ to 1 and $Norm$ normalizes the outgoing edges from each cell to sum to 1. The minimum edge weight is set to be 0.3 to ensure that neighbors in the original manifolds with no shared cross-species neighbors still retain connectivity: $\tilde{N}_{1,ij} = min\left(0.3, \tilde{N}_{1,ij}\right)$ and $\tilde{N}_{2,ij} = min\left(0.3, \tilde{N}_{2,ij}\right)$ for all edges $(i,j)$. We then scale the within-species edges from cell $i$ by the total weight of its cross-species edges: $\tilde{N}_{1,i} = \left(1 - \frac{1}{k}\sum_{j=1}^{n_2} C_{1,ij}\right)\tilde{N}_{1,i}$ and $\tilde{N}_{2,i} = \left(1 - \frac{1}{k}\sum_{j=1}^{n_1} C_{2,ij}\right)\tilde{N}_{2,i}$. Finally, the within- and cross-species graphs are stitched together to form the combined nearest neighbor graph $N$: $N = \left[\tilde{N}_1 \oplus C_1\right] \oplus \left[C_2 \oplus \tilde{N}_2\right]$. The overall alignment score between species 1 and 2 is defined as $S = \frac{1}{n_1 + n_2}\left(\sum_{i=1}^{n_1}\sum_{j=1}^{n_2} C_{1,ij} + \sum_{i=1}^{n_2}\sum_{j=1}^{n_1} C_{2,ij}\right)$.

## Homology graph refinement

### Update edge weights in the gene-gene bipartite graph with expression correlations

To compute correlations between gene pairs, we first transfer expressions from one species to the other: $\bar{Z}_{i,n_i m_j} = C_{i,n_i} Z_{j,m_j}$, where $\bar{Z}_{i,n_i m_j}$ is the imputed expressions of gene $m_j$ from species $j$ for cell $n_i$ in species $i$, and $C_{i,n_i}$ is row $n_i$ of the biadjacency matrix encoding the cross-species neighbors of cell $n_i$ in species $i$, all for $(i,j) = (1,2)$ and $(2,1)$. We similarly use the manifolds constructed by SAM to smooth the within-species gene expressions using kNN averaging: $\bar{Z}_{j,m_j} = N_{j,m_j} Z_{j,m_j}$, where $N_j$ is the nearest-neighbor graph for species $j$. We then concatenate the within- and cross-species gene expressions such that the expression of gene $m_j$ from species $j$ in both species is $\bar{Z}_{m_j} = \bar{Z}_{i,m_j} \oplus \bar{Z}_{j,m_j}$.

For all gene pairs in the initial unpruned homology graph, $G$, we compute their correlations, $G_{ab} := \theta(0) Corr(\bar{Z}_a, \bar{Z}_b)$, where $\theta(0)$ is a Heaviside step function centered at 0 to set negative correlations to zero. We then use the expression correlations to update the corresponding edge weights in $G$, which are again normalized through $G_{ab} = 0.5 + 0.5 tanh\left(10 G_{ab}/\max_b\left(G_{ab}\right) - 5\right)$.

## Annotation of cell atlases

To annotate the primary zebrafish and *Xenopus* cell types, the cell subtype annotations provided by the original publications (*Briggs et al., 2018*; *Wagner et al., 2018*) are coarsened using a combination of the manual matching and developmental hierarchies. For example, as 'involuting marginal zone' in *Xenopus* is manually matched to 'non-dorsal margin', 'dorsal margin' 'non-dorsal margin involuted', and 'dorsal margin involuted' in zebrafish, we label these cells as 'involuting marginal zone'. In cases where the matching is insufficient to coarsen the annotations, we use the provided developmental trees to name a group of terminal cell subtypes by their common ontogenic ancestor. Cell types that do not cluster well in the manifold reconstructed by SAM are excluded from the comparison. These include germline, heart, and olfactory placode cells, as they are mixed with other cell types in the *Xenopus* atlas. The germline cells are scattered across the reconstructed manifold and do not concentrate in a distinct cluster. The heart cells and olfactory placode cells are inextricably mixed with larger populations of intermediate mesoderm and placodal cells, respectively. Similarly, the iridoblast, epiphysis, *nanog*⁺, apoptotic-like, and forerunner cells are excluded because they do not cluster distinctly in the zebrafish atlas.

The annotations provided by their respective studies are used to label the cells in the *Spongilla*, *Hydra*, planarian, and mouse atlases. To annotate the schistosome cells, we use known marker genes to annotate the main schistosome tissue types (*Li et al., 2021*). Annotations for all single cells in all datasets are provided in *Supplementary file 1*.

## Visualization

The combined manifold $N$ is embedded into 2D projections using UMAP implemented in the scanpy package (*Wolf et al., 2018*) by *scanpy.tl.umap* with the parameter *min_dist* = 0.1. The sankeyD3 package (https://rdrr.io/github/fbreitwieser/sankeyD3/man/sankeyD3-package.html) in R is used to generate the sankey plots. Edge thickness corresponds to the alignment score between mapped cell types. The alignment score between cell types $a$ and $b$ is defined as $s_{ab} = \frac{1}{|c_a|+|c_b|}\left(\sum_{i\in c_a}\sum_{j\in c_b} C_{1,ij} + \sum_{i\in c_b}\sum_{j\in c_a} C_{2,ij}\right)$, where $c_a$ and $c_b$ are the set of cells in cell types $a$ and $b$, respectively. Cell type pairs with alignment score less than $z$ are filtered out. By default, $z$ is set to be 0.1.

The network graphs in *Figure 5C* are generated using the *networkx* package (https://networkx.github.io) in python. To focus on densely connected cell type groups, we filter out cell type pairs with alignment score less than 0.05.

## Identification of gene pairs that drive cell type mappings

We define $g_1$ and $g_2$ to contain SAMap-linked genes from species 1 and 2, respectively. Note that a gene may appear multiple times as SAMap allows for one-to-many homology. Let $X_{a_1 b_2}$ denote the set of all cells with cross species edges between cell types $a_1$ and $b_2$. We calculate the average standardized expression of all cells from species $i$ that are in $X_{a_1 b_2}$: $Y_{i,g_i} = \frac{1}{|\{x, x\in X_{a_1 b_2}\}|}\sum_{x\in X_{a_1 b_2}} \bar{Z}_{i,x,g_i}$, where

$\tilde{Z}_{i,x,g_i} \in \Re^{|g_i|}$ is the standardized expression of genes $g_i$ in cell $x$. The correlation between $Y_{1,g_1}$ and $Y_{2,g_2}$ can be written as $Corr(Y_{1,g_1}, Y_{2,g_2}) = \sum_{j=1}^{|g_1|} S(Y_{1,g_1})_j \circ S(Y_{2,g_2})_j$, where $S(Z)$ standardizes vector $Z$ to have zero mean and unit variance. We use the summand to identify gene pairs that contribute most positively to the correlation. We assign each gene pair a score: $h_g = T(S(Y_{1,g_1})) \circ T(S(Y_{2,g_2}))$, where $T(Z)$ sets negative values in vector $Z$ to zero in order to ignore lowly-expressed genes. To be inclusive, we begin with the top 1000 gene pairs according to $h_g$ and filter out gene pairs in which one or both of the genes are not differentially expressed in their respective cell types (p-value $> 10^{-2}$), have less than 0.2 SAM weight, or are expressed in fewer than 5% of the cells in the cluster. The differential expression of each gene in each cell type is calculated using the Wilcoxon rank-sum test implemented in the *scanpy* function *scanpy.tl.rank_genes_groups*.

## Orthology group assignment

We use the eggNOG mapper (v5.0) (*Huerta-Cepas et al., 2019*) to assign each gene to an orthology group with default parameters. For the zebrafish-to-*Xenopus* mapping, genes are considered orthologs if they map to the same vertebrate orthology group. For the pan-species analysis, we group genes from all species with overlapping orthology assignments. In *Figure 6B*, each column corresponds to one of these groups. As each group may contain multiple genes from each species, we present the expression of the gene with the highest enrichment score per species. All gene names and corresponding orthology groups are reported in *Supplementary file 5*.

## Paralog substitution analysis

SAMap outputs gene-gene correlations across the combined manifold for all pairs of genes in the homology graph. As determined by eggNOG, genes that map to the same orthology group for the two species' most recent common ancestor are considered orthologs, and those that map to the same orthology group more ancestral than Vertebrata are considered as paralogs. We note that as eggNOG does not provide an orthology group corresponding to the osteichtyan ancestor, our analysis does not include the paralogs that duplicated in between the osteichtyan and the vertebrate ancestors. If a gene has significantly higher correlation to one of its paralogs than its ortholog ($>0.3$ by default), we consider its ortholog to have been substituted. Paralog substitutions are identified using the *samap.analysis.ParalogSubstitutions* function provided by the SAMap package.

The evolutionary time period in which paralogs were duplicated can be inferred by identifying their most recent shared orthology group. We calculate the enrichment of paralog substitutions for each taxonomic level (i.e. Chordata, Bilateria, Metazoa, Opisthokonta, and Eukaryota) using the eggNOG orthology group assignments. We normalize the number of substituting paralogs by the total number of paralogs at each level to calculate the rate of paralog substitution across evolutionary time.

To quantify the enrichment of substituting paralogs in each cell type, we define a cell type-specific substitution score. We first assign paralog substitution events to cell types if the paralogous gene pairs are enriched in any of their mappings. Each cell type $k$ then has a set of substituting paralogs $P_k$. The score $S_k$ for cell type $k$ is calculated as $S_k = \sum_{i \in P_k} \frac{1-n_i}{m_k}$, where $n_i$ is the number of paralogs of ortholog $i$ normalized by the maximum number of paralogs observed across all genes to accounts for the fact that genes with more paralogs are more likely to match with substituting paralogs by random chance, and $m_k$ is the number of differentially expressed genes in cell type $k$. Similarly, the denominator accounts for the fact that cell types with more differentially expressed genes are more likely to have paralog substitutions by random chance. The substitution scores for cell types with annotated homologs across species are averaged.

## Phylogenetic reconstruction of gene trees

We generate gene trees to validate the identity of genes involved in putative examples of paralog substitution and of *Fox* and *Csrp* transcriptional regulators that are identified as enriched in contractile cells. For this, we first gather protein sequences from potential homologs using the eggnog version 5.0 orthology database (*Huerta-Cepas et al., 2019*). For the *Fox* and *Csrp* phylogenies, we include all Fox clade I (*Larroux et al., 2008*) and Csrp/Crip homologs, respectively, from the seven species included in our study.

Alignment of protein sequences is performed with Clustal Omega version 1.2.4 using default settings as implemented on the EMBL EBI web services platform (*Madeira et al., 2019*). Maximum likelihood tree reconstruction is performed using IQ-TREE version 1.6.12 (*Nguyen al., 2015*) with the ModelFinder Plus option (*Kalyaanamoorthy et al., 2017*). For the *Csrp* tree, we perform 1000 non-parametric bootstrap replicates to assess node support. For *Fox*, we utilize the ultrafast bootstrap support option with 1000 replicates. For each gene tree we choose the model that minimizes the Bayesian Information Criterion (BIC) score in ModelFinder. This results in selection of the following models: DCMut+R4 (*Csrp*) and VT+F + R5 (*Fox*). The final consensus trees are visualized and rendered using the ETE3 v3.1.1 python toolkit (*Huerta-Cepas et al., 2016*) and the Interactive Tree of Life v4 (*Letunic and Bork, 2019*).

## KOG functional annotation and enrichment analysis

Using the eggNOG mapper, KOG functional annotations are transferred to individual transcripts from their assigned orthology group. For enrichment analysis, all genes enriched in the set of cell type pairs of interest are lumped to form the target set for each species. For example, the target set for *Spongilla* archaeocytes used in *Figure 6C* is composed of all genes enriched between *Spongilla* archaeocytes and other invertebrate stem cells. Note that this set includes genes from other species that are linked by SAMap to the *Spongilla* archeocyte genes. We include genes from other species in the target set to account for differences in KOG functional annotation coverage between species. As such, the annotated transcripts from all seven species are combined to form the background set. We use a hypergeometric statistical test (*Eden et al., 2009*) to measure the enrichment of the KOG terms in the target genes compared to the background genes.

## Mapping zebrafish and *Xenopus* atlases using existing methods

For benchmarking, we use vertebrate orthologs as determined by eggNOG as input to Harmony (*Korsunsky et al., 2019*), LIGER (*Welch et al., 2019*), Seurat (*Stuart et al., 2019*), Scanorama (*Hie et al., 2019*), BBKNN (*Polański et al., 2019*), which are all run with default parameters. One-to-one orthologs are selected from one-to-many and many-to-many orthologs by using the bipartite maximum weight matching algorithm implemented in *networkx*. When using the one-to-one orthologs as input for SAMap, we run for only one iteration. The resulting integrated lower-dimensional coordinates (PCs for Seurat, Harmony, and Scanorama and non-negative matrix factorization coordinates for LIGER) and stitched graphs (BBKNN and SAMap) are all projected into 2D with UMAP (*Figure 2—figure supplement 1A*). The integrated coordinates are used to generate a nearest neighbor graph using the correlation distance metric, which is then used to compute the alignment scores in *Figure 2—figure supplement 1B*. The alignment scores for SAMap and BBKNN are directly computed from their combined graphs.

## In situ hybridization in schistosomes

*S. mansoni* (strain: NMRI) juveniles are retrieved from infected female Swiss Webster mice (NR-21963) at ~3 weeks post-infection by hepatic portal vein perfusion using 37°C DMEM supplemented with 5% heat inactivated FBS. The infected mice are provided by the NIAID Schistosomiasis Resource Center for distribution through BEI Resources, NIH-NIAID Contract HHSN272201000005I. In adherence to the Animal Welfare Act and the Public Health Service Policy on Humane Care and Use of Laboratory Animals, all experiments with and care of mice are performed in accordance with protocols approved by the Institutional Animal Care and Use Committees (IACUC) of Stanford University (protocol approval number 30366). In situ hybridization experiments are performed as described previously (*Tarashansky et al., 2019*), using riboprobes synthesized from gene fragments cloned with the listed primers: collagen (Smp_170340): GGTGAAGAAGGCTGTTGTGG, ACGA TCCCCTTTCACTCCTG; tropomyosin (Smp_031770): AAGCTGAAGTCGCCTCACTA, CATATGCCTC TTCACGCTGG; troponin (Smp_018250): CGTAAACCTGGTCAGAAGCG, ATCCTTTTCCTCCA-GAGCGT; myosin regulatory light chain (Smp_132670): GAGACAGCGAGTAGTGGAGG, TGCCTTC TTTGATTGGAGCT; wnt11 (Smp_156540): TGTGGTGATGAAGATGGCAG, CCACGGCCACAA-CACATATT; frizzled (Smp_174350): CGAACAGGCGCATGACAATA, TGCTAGTCCTGTTGTCGTGT.

## Acknowledgements

We thank D Wagner and C Juliano for sharing data and essential discussions. We also thank S Granick, L Luo, and J Kebschull for their critical reading of the manuscript. AJT is a Bio-X Stanford Interdisciplinary Graduate Fellow. JM and DA thank the support from an Advanced grant of the European Commission ('NeuralCellTypeEvo' 788921). This work is supported by a Beckman Young Investigator Award and an NIH grant (1R35GM138061) to BW.

## Additional information

### Funding

| Funder | Grant reference number | Author |
| --- | --- | --- |
| Arnold and Mabel Beckman Foundation | Beckman Young Investigator Award | Bo Wang |
| National Institutes of Health | 1R35GM138061 | Bo Wang |
| European Commission | 788921 | Jacob M Musser Detlev Arendt |

The funders had no role in study design, data collection and interpretation, or the decision to submit the work for publication.

### Author contributions

Alexander J Tarashansky, Conceptualization, Data curation, Software, Formal analysis, Validation, Investigation, Visualization, Methodology, Writing - original draft, Writing - review and editing; Jacob M Musser, Data curation, Formal analysis, Investigation, Writing - review and editing; Margarita Khariton, Visualization, Writing - review and editing; Pengyang Li, Investigation; Detlev Arendt, Stephen R Quake, Supervision, Writing - review and editing; Bo Wang, Conceptualization, Supervision, Investigation, Writing - original draft, Writing - review and editing, Project administration

### Author ORCIDs

Detlev Arendt (iD) http://orcid.org/0000-0001-7833-050X
Bo Wang (iD) https://orcid.org/0000-0001-8880-1432

### Ethics

Animal experimentation: All experiments with and care of mice are performed in accordance with protocols approved by the Institutional Animal Care and Use Committees (IACUC) of Stanford University (protocol approval number 30366).

### Decision letter and Author response

Decision letter https://doi.org/10.7554/eLife.66747.sa1
Author response https://doi.org/10.7554/eLife.66747.sa2

## Additional files

### Supplementary files

• Supplementary file 1. Cell atlas metadata and cell annotations. Metadata include the number of cells, number of transcripts in the transcriptome, median number of transcripts detected per cell, the reference transcriptome used in this study, database through which the transcriptomes are provided, technology used for constructing the cell atlases, atlas data accessions, processing notes, and references. Leiden clusters and cell type annotations are reported for cells in each atlas. The Zebrafish and *Xenopus* tables include both the original cell type annotations and those used in this study. *D. rerio*, *X. tropicalis*, and mouse annotations include developmental stages.

• Supplementary file 2. Cell type annotations for the zebrafish-*Xenopus* mapping. Correspondence between the cell type annotations provided in the original study (*Briggs et al., 2018*; *Wagner et al.,*

*2018*) and corresponding annotations used in this study is provided for both *D. rerio* and *X. tropicalis* atlases.

• Supplementary file 3. Mapping of zebrafish-*Xenopus* atlases with individual cell types removed. The two highest-scoring partners are reported for each cell type in the original mapping and the mapping after its homolog was removed. The new mappings are categorized as being present in the original analysis, not present in the original analysis but connecting developmentally related cells, or neither.

• Supplementary file 4. Identified paralogs with greater expression similarity than orthologs in the zebrafish-*Xenopus* mapping. Each row contains a pair of vertebrate-orthologous genes and a corresponding pair of eukaryotic paralogs with higher correlation in expression compared to the orthologs, the expression correlations for ortholog and paralog pairs, the difference between their correlations, the paralogs' last common ancestor, and the cell types in which the genes are enriched. Highlighted rows are shown in *Figure 3A*.

• Supplementary file 5. Genes enriched in contractile cell types and invertebrate stem cells highlighted in *Figure 6*. The IDs of the genes enriched in the contractile and invertebrate stem cell types are provided along with the IDs of the eggNOG orthology groups to which they belong. In cases where multiple genes from a species belonging to the same orthology group are enriched, the most differentially expressed gene is shown. The descriptions in the stem cell table are orthology annotations associated with the *Spongilla* genes provided in the original study (*Musser et al., 2019*).

• Supplementary file 6. Transcript IDs corresponding to the multipotent stem cell enriched genes shown in *Figure 6D*.

• Supplementary file 7. Cell types in the cell type families shown in *Figure 5C*. For the schistosome cell types, we annotate two neural clusters, both of which express the neural marker *complexin* (*Li et al., 2021*). One of the clusters expresses the antigen *SmKK7*, so we label the clusters 'Neural' and 'Neural_KK7', respectively. The 'Muscle' population contains non-stem cells expressing *troponin*. The 'Tegument_prog' and 'Tegument' populations consist of cells expressing tegument progenitor and differentiated marker genes, respectively, as reported in a previous study (*Wendt et al., 2018*).

• Transparent reporting form

## Data availability

All data analyzed during this study are available through various sources as listed in Supplementary file 1.

The following previously published datasets were used:

| Author(s) | Year | Dataset title | Dataset URL | Database and Identifier |
|---|---|---|---|---|
| Musser JM | 2019 | Whole-body single-cell RNA sequencing reveals components of elementary neural circuits in a sponge | https://www.ncbi.nlm.nih.gov/geo/query/acc.cgi?acc=GSE134912 | NCBI Gene Expression Omnibus, GSE134912 |
| Siebert S, Cazet J, Farrell JA | 2018 | Stem cell differentiation trajectories in *Hydra* resolved at single cell resolution | https://www.ncbi.nlm.nih.gov/geo/query/acc.cgi?acc=GSE121617 | NCBI Gene Expression Omnibus, GSE121617 |
| Fincher CT, Wurtzel O, de Hoog T, Kravarik KM, Reddien PW | 2018 | Cell type transcriptome atlas for the planarian *Schmidtea mediterranea* | https://www.ncbi.nlm.nih.gov/geo/query/acc.cgi?acc=GSE111764 | NCBI Gene Expression Omnibus, GSE111764 |
| Zeng A, Li H, Sánchez Alvarado A | 2017 | Tetraspanin family member functionally resolves and facilitates the purification of adult pluripotent stem cells used for whole-body regeneration | https://www.ncbi.nlm.nih.gov/geo/query/acc.cgi?acc=GSE107873 | NCBI Gene Expression Omnibus, GSE107873 |

| Xue Y, Li P, Quake SR, Wang B | 2020 | Single-cell analysis reveals regulation of germline stem cell fate in the human parasite *Schistosoma mansoni* | https://www.ncbi.nlm.nih.gov/geo/query/acc.cgi?acc=GSE147355 | NCBI Gene Expression Omnibus, GSE147355 |
|---|---|---|---|---|
| Wagner DE, Weinreb C, Collins ZM, Megason SG, Klein AM | 2018 | Systematic mapping of cell state trajectories, cell lineage, and perturbations in the zebrafish embryo using single cell transcriptomics | https://www.ncbi.nlm.nih.gov/geo/query/acc.cgi?acc=GSE112294 | NCBI Gene Expression Omnibus, GSE112294 |
| Briggs JA, Weinreb C, Wagner DE, Megason S, Peshkin L, Kirschner MW, Klein AM | 2018 | The dynamics of gene expression in vertebrate embryogenesis at single cell resolution | https://www.ncbi.nlm.nih.gov/geo/query/acc.cgi?acc=GSE113074 | NCBI Gene Expression Omnibus, GSE113074 |
| Griffiths J | 2018 | Embryo Timecourse | https://github.com/MarioniLab/EmbryoTimecourse2018 | Github, ab59525 |

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
