## [Decision Letter]

**Acceptance summary:**

The development of single-cell genomic methods has transformed our understanding of cell types and their attributes across organisms. Here, Tarashansky et al. develop SAMap (Self-Assembling Manifold mapping), a graph-based data integration method which builds upon their previously described SAM algorithm, to facilitate assignment of homologous genes and cell types across diverse species. As the authors show, this empowers comparative analyses across phyla to facilitate cellular annotation and examine the evolutionary origins of cellular diversity. Overall, the algorithm has the potential to be broadly enabling for comparative cellular atlasing.

**Decision letter after peer review:**

Thank you for submitting your article "Mapping single-cell atlases throughout Metazoa unravels cell type evolution" for consideration by *eLife*. Your article has been reviewed by 3 peer reviewers, and the evaluation has been overseen by Alex K Shalek as the Reviewing Editor and Naama Barkai as the Senior Editor. The reviewers have opted to remain anonymous.

The Reviewing Editor has drafted this to help you prepare a revised submission.

Essential revisions:

1. While authors clearly demonstrate the promise of SAMap, the manuscript would benefit from an accessible discussion of the algorithm's potential applications, limitations and drawbacks to help inform use. For example, how does the algorithm depend on cell numbers, data quality, or the use of a consistent experimental method? If a cell type is missing from one atlas (e.g., due to limited cell numbers), will the algorithm overfit? Performing downsampling analyses, leaving one cluster out (e.g., when comparing zebrafish and *Xenopus* (Figure 2)), or linking datasets across methods (e.g., Smart-Seq2 and 10x; inDrop and 10x) would help to address these points.

2. The authors' analyses present several intriguing evolutionary observations such as those on widespread paralog substitution, the multifunctionality of ancestral contractile cells, and the existence of a deeply conserved gene module associated with multipotency. Each would benefit from further investigation. For example, with respect to the paralogs, are similar levels of substitution observed when paralogs are excluded during manifold assembly (i.e., do they drive cell type assignments)? Similarly, how does paralog substitution depend on how recently those paralogs arose or their stability? Meanwhile, the points on multifunctionality and multipotency would benefit from deeper analysis and discussion, or more cautious language. Re: the first point above, each observation would also benefit from presentation of potential alternative interpretations in the Discussion section.

*Reviewer #1 (Recommendations for the authors):*

I am very supportive of this manuscript and agree with the authors assessment of the utility of the method they have developed. My major concerns emerge from some of the evolutionary interpretations of the results. In particular, I wonder whether it would be possible to exclude the paralog pairs for which substitution has been observed during manifold assembly to determine whether those paralogs are driving cell type assignments leading to a tautology. I would recommend that in the three instances where evolutionary conclusions are proposed, the authors consider alternative interpretations within their discussion.

*Reviewer #2 (Recommendations for the authors):*

The paper is solid.

*Reviewer #3 (Recommendations for the authors):*

The study and methods give a great conceptual overview of the novel approach, but the details for implementation are not clear, and the github is not well documented. I would encourage further details and more clear documentation on the github – for example the paralog substitution findings are an important result and use case, but there is limited methods description and it is unclear how to run the function.

Finally, the reciprocal BLAST is slow to run, especially for all by all transcripts, but it only needs to run once. I would consider posting the results of this analysis on the github for widely used species pairs, which could also accelerate adoption by reducing the barrier to running the full suite.

---

## [Author Response]

Essential revisions:1. While authors clearly demonstrate the promise of SAMap, the manuscript would benefit from an accessible discussion of the algorithm's potential applications, limitations and drawbacks to help inform use. For example, how does the algorithm depend on cell numbers, data quality, or the use of a consistent experimental method? If a cell type is missing from one atlas (e.g., due to limited cell numbers), will the algorithm overfit? Performing downsampling analyses, leaving one cluster out (e.g., when comparing zebrafish and *Xenopus* (Figure 2)), or linking datasets across methods (e.g., Smart-Seq2 and 10x; inDrop and 10x) would help to address these points.

1.1. Thanks for raising these important questions. Now we discuss the technical specifics of SAMap more explicitly (lines 133-141) and included a new figure (Figure 1—figure supplement 1). The datasets analyzed in this study range from thousands to hundreds of thousands of cells sequenced using different methods. For example, the planarian atlas contained 50,000 cells sampled with Drop-Seq and the schistosome atlas contains 7,000 cells sampled with Smart-Seq2. In addition, we have recently applied SAMap to map a dataset of lower quality collected from *Amphimedon queenslandica* (Sebé-Pedrós et al., 2018) using MARS-seq to the 10x dataset of *Spongilla lacustris (Musser et al., 2019).* We found broad concordance between the major cell type families (see Author response image 1). This newer result is included in another paper that is currently under review. SAMap’s use of mutual connectivity to determine alignment strength rather than absolute measures of similarity such as correlation makes it robust to technical batch effects such as those related to the different library preparation methods used.

Although SAMap scales well to the majority of currently available whole-organism cell atlases, datasets will continue to get larger. Currently, the most memory-intensive steps in SAMap are the neighborhood coarsening of cross-species edges and the cross-species gene expression imputation to calculate gene-gene correlations, both of which can be intractable for datasets containing millions of cells. Our current solution is to chunk these operations into smaller blocks for large datasets to avoid memory limitations, but the runtime increases significantly as a result. An option for mapping massive datasets may be to downsample each atlas. So long as all cell types are preserved and remain separable during downsampling, we expect the mapping results to be the same. We now discuss these points in the text (lines 409-418).

1.2. To evaluate if SAMap overfits in cases where some cell types are missing, we performed dropout experiments in which we systematically removed each cell type that has an annotated homolog in the comparison of zebrafish and frog atlases. Cell types whose homologous partners were removed weakly mapped to closely related cell types, and many of these links were already present in the original analysis (Supplementary File 3). For example, optic cells from both species are also connected to eye primordium, frog skeletal muscles to zebrafish presomitic mesoderm, and frog hindbrain to zebrafish forebrain/midbrain. While we observed several mappings that were not present in the original analysis, their alignment scores were all barely above the detection threshold of SAMap. Moreover, most of these edges were mapped between cell types with similar developmental origins, with the only exception being the zebrafish neural crest mapped to the frog otic placode in the absence of frog neural crest cells. Examining the genes that support this mapping revealed that both cell types express *sox9* and *sox10*, two TFs previously implicated to form a conserved gene regulatory circuit common to otic/neural crest cells (Betancur et al., 2011). These results are now discussed in the text (lines 194-210).

2. The authors' analyses present several intriguing evolutionary observations such as those on widespread paralog substitution, the multifunctionality of ancestral contractile cells, and the existence of a deeply conserved gene module associated with multipotency. Each would benefit from further investigation. For example, with respect to the paralogs, are similar levels of substitution observed when paralogs are excluded during manifold assembly (i.e., do they drive cell type assignments)? Similarly, how does paralog substitution depend on how recently those paralogs arose or their stability? Meanwhile, the points on multifunctionality and multipotency would benefit from deeper analysis and discussion, or more cautious language. Re: the first point above, each observation would also benefit from presentation of potential alternative interpretations in the Discussion section.

2.1. Thanks for these great suggestions. SAMap yields a similar combined manifold when using only one-to-one orthologs (Figure 2E), suggesting that at least for the zebrafish-frog comparison the paralogs are not driving the manifold mapping. To rule out the possibility that these paralogs were linked spuriously during the homology refinement steps of SAMap, we repeated the paralog substitution analysis on the combined manifold constructed using only one-to-one orthologs. This identified a largely similar set of paralog substitution events, although weaker manifold alignment when restricting the mapping to one-to-one orthologs led to the loss of some substitution paralogs that showed lower correlations. These new results are now reported in Figure 3—figure supplement 1 and discussed in the text (lines 242-251).

2.2. To determine whether paralog substitution depends on how recently they arose, we used the orthology groups provided by Eggnog to infer when paralogs duplicated during evolution. We found that more recent paralogs substitute at higher rates than more ancestral paralogs, which is in line with the expectation that less diverged genes are likely more capable of functionally substituting each other (Figure 3C). We also used the paralog substitution score to quantify the rate of paralog substitution in each cell type and observed that substituting paralogs are expressed in a wide variety of cell types, with some (e.g., dorsal organizer) exhibiting higher rates than others (Figure 3B), indicating uneven diversification rates of paralogs across cell types. Unfortunately, assessing the stability of paralog substitutions within a clade requires more cell atlases than what are available at the moment. This analysis needs to densely sample species within clades and at key branching points along the tree of life. We now discuss these new results and possible future directions in the text (lines 229-231, lines 237-242, and lines 448-455).

2.3. We apologize for the confusing statement on muscle cell type functions. We have modified the text (lines 356-359) to clarify that ancestral contractile cells may already possess the broad assemblage of gene modules associated with different functional aspects of modern muscle cell types, including the adhesion complex that connects cells, actomyosin networks that drive contractility, and signaling pathways that stimulate contraction.

2.4. To expand the analysis on multipotency, we extend the comparison to include multipotent stem cells (MSCs), lineage-restricted stem cells, and differentiated cells for all four invertebrates analyzed in this study. Importantly, this new analysis identified several transcription factors and chromatin modifiers enriched in MSCs that may play essential roles in establishing gene expression programs associated with multipotency (Figure 6D). These new results are nor discussed in the text (lines 382-399). Thanks for this great suggestion.

Reviewer #1 (Recommendations for the authors):I am very supportive of this manuscript and agree with the authors assessment of the utility of the method they have developed. My major concerns emerge from some of the evolutionary interpretations of the results. In particular, I wonder whether it would be possible to exclude the paralog pairs for which substitution has been observed during manifold assembly to determine whether those paralogs are driving cell type assignments leading to a tautology. I would recommend that in the three instances where evolutionary conclusions are proposed, the authors consider alternative interpretations within their discussion.

Thanks for the support and great suggestions! We now have strengthened these parts through new analysis and better discussions. Please see reply to editor’s comments, 2.1-2.4.

Reviewer #3 (Recommendations for the authors):The study and methods give a great conceptual overview of the novel approach, but the details for implementation are not clear, and the github is not well documented. I would encourage further details and more clear documentation on the github – for example the paralog substitution findings are an important result and use case, but there is limited methods description and it is unclear how to run the function.

We have made a number of improvements to the Github for aid usability, summarized here:

1. We now provide a Docker container which launches a Jupyter notebook server configured to run SAMap and all its associated functions.

2. The paralog substitution analysis function has better documentation and accepts both cross-species and within-species paralogs. Within-species paralogs are translated to cross-species paralogs automatically by using the cross-species orthologs as anchors.

3. We now provide a built-in function to convert Eggnog mapping tables to ortholog gene pairs at specified taxonomic levels.

4. We provided a built-in function to perform functional enrichment analysis given any functional annotations (e.g. GO terms or KOG annotations).

5. We provide several functions to facilitate convenient visualization of SAMap results:

a. A function that launches an interactive GUI provided by the SAM package to facilitate interactive exploration of the SAMap manifold.

b. A function that creates an interactive Sankey plot.

c. A function that displays a cell type mapping heatmap.

d. A function to overlay expression patterns on the combined manifold to create plots like the ones shown in Figure 3A.

e. A function to create the enrichment plots shown in Figure 6A,C.

6. All exposed functions now have docstrings with descriptions for each input parameter and returned output.

7. We have provided more thorough tutorials in a Jupyter notebook on Github, along with a Jupyter notebook showing all possible outputs of a complete run.

We have also added a section to the methods describing the paralog substitution analysis in more detail (lines 692-721).

Finally, the reciprocal BLAST is slow to run, especially for all by all transcripts, but it only needs to run once. I would consider posting the results of this analysis on the github for widely used species pairs, which could also accelerate adoption by reducing the barrier to running the full suite.

This is an excellent suggestion. We have added a database of species codes, transcriptome versions, and corresponding BLAST tables to the Github. In the future, we aim to release an online database and interactive explorer that documents mappings (and all associated inputs/outputs) between all pairs of species uploaded to the database.